# Evolutionary echoes of emotion: Humans mimic other primate expressions

Ursula Hess[1]*, Till Kastendieck[1], M. Gizem Erkol[1], Heidi Mauersberger[1],
Marina Davila-Ross[2], Katja Liebal[3], Elisabetta Palagi[4]

1 Department of Psychology, Humboldt University of Berlin; Berlin, Germany, 2 School of Psychology, Sport, and Health Sciences, University of Portsmouth, Portsmouth, United Kingdom, 3 Human Biology and Primate Cognition, Leipzig University, Leipzig, Germany, 4 Department of Biology, University of Pisa, Pisa, Italy

☙ These authors contributed equally to this work.
* Ursula.Hess@hu-berlin.de

## Abstract

Humans readily mimic the emotional behavior of conspecifics -- a behavior linked to empathy. Yet, whether humans unconsciously mimic the emotional expressions of non-human primates remains an open question. Human observers watched short videos with positive (play-face), negative (open-mouth threat display) or neutral expressions by monkeys and apes (while their own facial expressions were filmed and automatically coded), rated the expressions for emotional content and indicated their degree of liking of and closeness to the primates. Participants mimicked both positive and negative expressions and were able to correctly identify the expressions as positive or negative. These findings shed new light on the deep-rooted, cross-species nature of emotional connection, suggesting that humans are able to empathize and mirror the emotions of other species.

## 1. Introduction

The matching or mimicry of the emotional behavior of conspecifics is prevalent in many species. Next to humans [1] it has, for example, been observed in orangutans [2], geladas [3] and some macaque species [4,5], but also sun bears [6], dogs [7] and meerkats [8]. In humans, but also in non-human primates, the imitation of emotional behavior – emotional mimicry – has been shown to be relevant for the establishment of affiliative relations [1,9]. Specifically, mimicry is indicative of an affiliative stance towards the mimickee [1,10] and is often considered a "low road" to empathy [11], as such it is one important facet of human empathy.

There is good evidence that infant and juvenile non-human primates spontaneously mimic human facial gestures, such as tongue protrusion and lip smacking [(see, e.g., 12, 13)]. Moreover, humans and chimpanzees have been found to engage in voluntary imitation of each other's behavior [14]. Notably also, capuchin monkeys

**Data availability statement:** The data files and markdown are available at [https://osf.io/n4cdj/files/osfstorage].

**Funding:** The author(s) received no specific funding for this work.

**Competing interests:** The authors have declared that no competing interests exist.

show more affiliation towards humans who were instructed to imitate than when no imitation took place [10], suggesting that mimicry can serve to signal affiliation across species.

Curiously however, there are to date no studies assessing whether humans spontaneously mimic the emotional expressions of non-human primates. The goal of the present research was to close this gap.

Given the importance of mimicry behavior for affiliative relations between individuals [1,15], the question of whether such behavior is shown across species is intriguing. If humans imitate the emotional behavior of non-human primates, this would be suggestive for humans' capacity for empathic reactions towards them [11].

According to the emotional mimicry as social regulator view [1,15], humans should mimic the expressions of others, including non-humans, provided they are able to understand them and they feel a certain level of closeness to the expresser. In human-human interaction, this level of closeness is almost a default setting, such that for the most part, humans mimic fellow humans unless there is an explicit reason for reduced closeness [16,17]. In fact, humans mimic not only human-like avatars [17] but also crude drawings of human emotional faces, provided the expressions are perceived as emotionally relevant [18].

Thus, two factors seem relevant. First, are humans able to recognize the valence of the expression, which seems to suffice for mimicry [15] and, second, do they feel sufficiently close to non-human primates for mimicry to occur.

In this context, it is important to emphasize that we use an observer focused perspective when using the terms emotional mimicry and emotional expression. It is difficult if not impossible to assert what any organism, humans and other primates included, experiences when showing a given expression. However, humans nonetheless not only attribute emotions to such expressions [18] but also act accordingly [19]. Thus, to the human observer the primate shows an emotion and the observer will mimic (or not) the expression in line with this assumption. Thus, human empathy with a primate can only be based on the human observers'attributions not on ground truth knowledge about inner states.

Some research indicates that humans report more empathy towards animals such as non-human primates (that are more closely related to humans) than for example towards birds and reptiles [19]. Further, humans are generally able to recognize emotional expressions of other animal taxa, such as cats [20] and dogs [21]. Depending on the emotion, humans are better able to detect emotions in our closest relatives (chimpanzees/bonobos) than in dogs, with anger being generally well detected across species [22]. Human listeners are also to some degree able to identify emotions of vocalizations by rhesus macaques [23] as well as primate facial expressions, even though the level of accuracy varies with primate species and experience [24]. Importantly, primate and human emotion expressions, despite various differences, share many features [25,26] and hence an ability to rate expressions as positive or negative is plausible. This notion is also supported by a study showing that adult humans rate complex scenes with multiple protagonists who are either human or bonobos similarly with regard to the valence and arousal of the picture [27]. Even

though the focus here was not on individual facial expressions per se, the expressive behavior of the bonobos played a role, as children tended to misidentify pictures including silent bared-teeth displays as positive [27]. Yet, in a task where attentional bias to emotional versus neutral expressions was assessed, the bias shown for human expressions was not found for images showing chimpanzee expression suggesting a lack of relevant expertise [28].

The present research is the first to address whether humans who have little or no experience with non-human primates, can not only recognize the emotions associated with non-human primate facial expressions but also spontaneously mimic these expressions. In an online experiment, participants saw videos of positive (play-face: mouth is open and relaxed with rounded lip corners, teeth are visible), negative (open-mouth threat display: mouth is pulled open, teeth are prominently visible) and neutral expressions (mouth is closed and relaxed). Fig 1 shows example stills of the expressions.

Following each video, participants rated both the degree of positivity/negativity of the expression and the levels of different discrete emotions (anger, disgust, fear, happiness, sadness, surprise) as well as perceived psychological closeness and liking towards the expresser. While they were watching the video, observers' facial expressions were filmed using their webcam. The expressions were FACS [26] coded using OpenFace 2.0 [28]. This method allows a detailed description of the participants' facial activity in terms of facial action units (AUs). Expressions with high intensity of AU04 (frown) and low intensity of AU12 (pulls the lips up) and AU06 (wrinkles around the eyes) index negative facial expressions, whereas the reverse pattern indexes positive expressions [29].

The videos showed the animals in natural surroundings. As such, most of the upper body and, in some cases, the whole body of the individuals was visible. That is, emotion cues were not limited to the face. However, there is good evidence that humans engage in facial mimicry even when the emotional signal is not presented via facial expressions [29,27] as long as they understand the emotion being signaled [30].

## 2. Methods

### Participants

A power simulation using simr [31] assuming an effect size of d = .32 (based on data from previous online studies involving facial mimicry) for the focal mimicry analysis suggested 200 participants for power > 90% with an alpha of .05 for the focal analyses (see power curve in the supporting information). A total of 212 (103 women, 107 men, 2 gender unknown) participants with a mean age of 40 years (SD = 13) completed the task and provided usable video material. Data from an additional 18 participants were excluded due to technical problems or noncompliance with the video instructions (e.g., eating during the experiment, face not visible on camera).

Participants were recruited via Prolific Academic and were paid €3.11 for the 20-minute task (which is classified as "good" payment in Prolific). The study was approved by the Institutional Review Board of the Department of Psychology

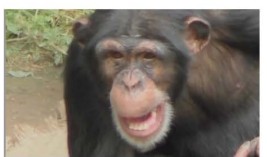
Positive (play face)

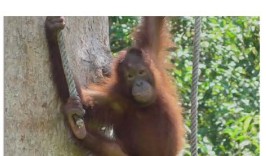
Neutral

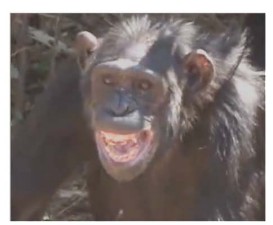
Negative (open-
mouth threat display)

**Fig 1. Examples for positive, neutral and negative expressions.** Copyright held by the authors.

of Humboldt University of Berlin and conducted in accordance with the Declaration of Helsinki and the German Research Foundation's Guidelines for Safeguarding Good Research Practice. A master's thesis that was part of the larger project was preregistered via OSF (https://osf.io/9rw2m). The data reported herein are not part of the thesis. Data were collected during February 2025.

## Measures

**Dimensional Emotion Perception Ratings**. Participants rated the positive and negative valence of each non-human primate expression on two Likert scales (1 = not at all, 7 = very much).

   **Discrete Emotion Profile**: Participants rated the degree to which the non-human primate seemed to express each of seven emotions (happiness, anger, sadness, fear, disgust, and surprise) using a 7-point scale ranging from 1 – not at all to 7 – very much. The emotion profile was used to assess how human observers interpret the expressions using emotion terms with which they are familiar.

   **Perceived closeness and liking**. Following the emotion ratings, participants were asked to rate their *closeness* to the primate shown using the Inclusion of Other in the Self-scale [32]. For this, they used a slider that moved two circles that represented themselves and the primate to a distance that represented their perceived closeness (1 – most distant, 101 –closely overlapping). They also indicated their *liking* of the primate on a 7-point scale (1- not at all, 7 – very much). See below for additional measures not reported here.

   **Emotional Mimicry**: Emotional mimicry was assessed as using OpenFace 2.0 [33], an open-source tool for facial behavior analysis. The software tracks facial activity based on the Facial Action Coding System [34]. We extracted information for AU4 (brow furrow), AU6 (cheek raise), and AU12 (lip corner pull). Mimicry of negative expressions is indexed by a relatively stronger activity of AU4 compared to AU6 and AU12; the reverse pattern indexes mimicry of positive expressions [35].

## Video stimuli

Stimuli consisted of videos showing apes (eight chimpanzees, two orangutans, two gorillas, and one bonobo) and monkeys (one langur and one crested macaque) who showed either a positive (play face), neutral or negative (open-mouth threat display) expression. For each expression five different primates were shown. The videos were sourced from a pool of several primate research archives provided by Elisabetta Palagi, Marina Davila-Ross, and Katja Liebal, as well as primate documentaries (e.g., Netflix, BBC, NatGeo). Videos were selected based on image quality, duration, and the visibility of expressions. Only videos showing a single animal in the scene were included. The preselected videos were independently coded by Elisabetta Palagi and Marina Davila-Ross, with regard to the expression shown. For the final video selection, 100% agreement regarding the expression was obtained. Videos varied in length from 5.7 to 6.5 seconds.

## Procedure

Participants were informed about the content and length of the task. Furthermore, they were informed that participation in the experiment is only possible if they have a webcam enabled computer/laptop and agree to a webcam recording of their face during the experiment. Informed consent included standard details on compensation, confidentiality, and contact information as well as detailed information on (video) data storage and processing. Participants who gave informed consent were instructed to set up their webcam to allow recording, to arrange sufficient lighting, and to refrain from eating or covering their face during the experiment.

   Participants then provided general socio-demographic information. Before beginning the main experiment, participants completed a single practice trial with a non-human primate video displaying a pant hoot expression to familiarize themselves with the task.

During each trial, participants saw the videos in random order. Videos were preceded by a 1.5 sec fixation cross (on grey screen) and followed by a 1.5 sec grey screen. While they watched the video, participants' faces were recorded. Following each video, participants completed the rating scales. After completing the video trials, they provided global ratings of their perceived connectedness to nature, closeness to primates, and phylogenetic closeness to specific primate species (these scales are not part of the present report). At the end of the experiment, participants were fully debriefed about the study's goals and were provided with a code to receive compensation via Prolific.

The research was preregistered [https://osf.io/9rw2m] and received Ethics approval from the Departmental Ethics board of the Department of Psychology at Humboldt-University of Berlin [Addendum to Proposal #2020−39]. The complete data as well as a markdown for all reported analyses as well as supplementary analyses is available and can be found in S1 R Markdown. The data files and markdown are available at [https://osf.io/n4cdj/files/osfstorage].

## Results

Data analyses were conducted using LMM with sum coded contrasts for Expression and Helmert contrast for AUs (lme4, *32,* and lmerTest, *33*). Post-hoc tests were conducted using emmeans [36–38]. A markdown for all analyses can be found in S1 R Markdown.

### Emotion recognition

We first assessed whether participants recognized the expressions. An LMM with the random factor participants revealed a significant effect of Expression for ratings of both Positivity, $F(2, 1977) = 81.44$, $p < .001$, $\eta_p^2 = .08$, and Negativity of the Expression, $F(2, 1979) = 148.01$, $p < .001$, $\eta_p^2 = .13$. Participants rated positive expressions as significantly more positive ($M = 3.48$, $SD = 1.11$, $CI_{95}[3.33, 3.64]$) and less negative ($M = 2.66$, $SD = 1.02$, $CI_{95}[2.52, 2.80]$) than negative expressions ($M_{pos} = 2.42$, $SD = 1.10$, $CI_{95}[2.26, 2.57]$, $t_{1977} = 12.11$, $p < .001$; $M_{neg} = 4.12$, $SD = 1.31$, $CI_{95}[3.94, 4.30]$, $t_{1977} = 15.65$, $p < .001$). Neutral expressions ($M_{pos} = 2.69$, $SD = 1.12$, $CI_{95}[2.53, 2.84]$; $M_{neg} = 2.79$, $SD = 1.23$, $CI_{95}[2.62, 2.86]$) were rated as more positive than negative pressions, $t_{1978} = 2.63$, $p = .024$, but less positive than positive expressions, $t_{1977} = 9.49$, $p < .001$. Even though neutral expressions were rated as less negative than negative expression, $t_{1977} = 14.01$, $p < .001$, they did not differ in negativity from positive expressions, $t_{1976} = 1.53$, $p = .275$.

Fig 2 shows the ratings on the discrete emotion scales. Positive expressions were rated as predominantly happy, negative expressions as predominantly angry and neutral expressions as predominantly sad (for more details, post-hoc tests and a table with the summary statistics see S1 File).

Overall, these data suggest that participants were rather good at assessing whether an expression signals positive versus negative affect or neutrality. They also associated the negative expressions more with anger (but also with fear and surprise) and the positive expressions more with happiness. Neutral expressions were rated somewhat sadder, possibly due to the low arousal in these expressions.

### Mimicry

Mimicry was indexed by a relatively higher level of AU4 (frown) compared to AU6 (wrinkles around the eyes) and AU12 (lip corners pulled up) for negative expressions and the reverse pattern for positive expressions. Fig 3 shows the means and standard errors for these action units.

The observed pattern is congruent with the notion that participants mimic the valence of the expressions. A linear model on the within participant z-transformed AU data, with the factors AU and Expression reveals a significant effect of expression which was qualified by a significant interaction between AU and Expression, $F(4, 7750) = 5.18$, $p < .001$, $\eta_p^2 = .003$, indicating that different AU patterns were observed in response to different primate expressions.

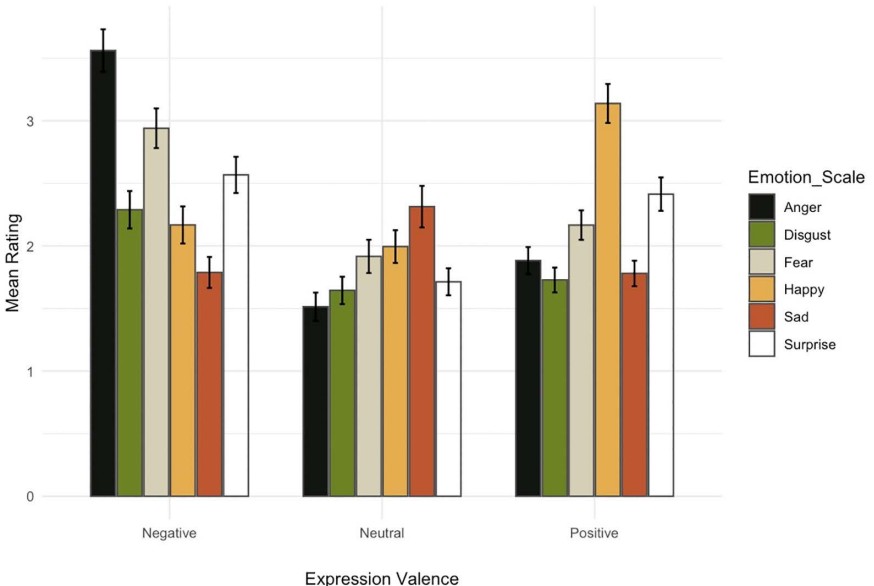

**Fig 2. Mean ratings for discrete emotions as a function of expression valence.**

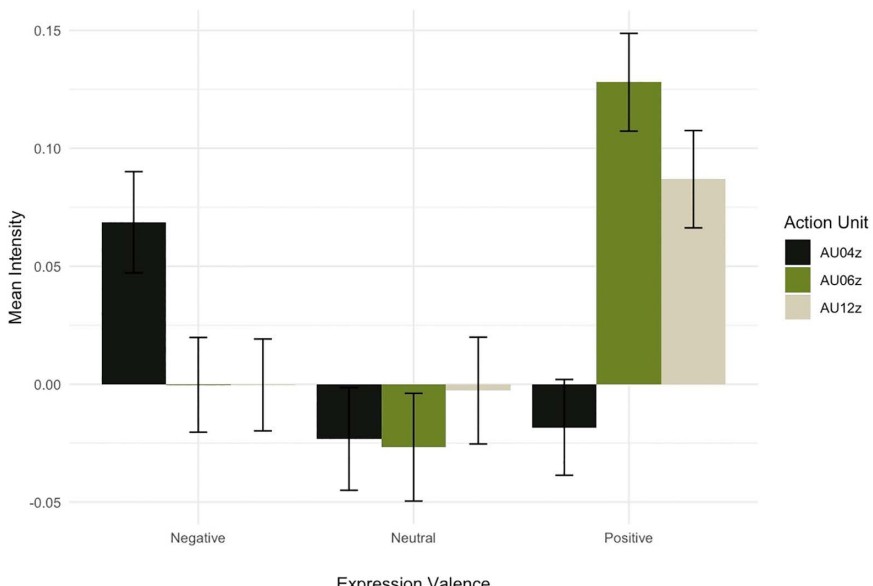

**Fig 3. Mean intensity (z-scores) of AU04, AU06, and AU12 as a function of expression valence.**

Post-hoc tests using a Helmert contrast to compare AU04 to the mean of AU06 and AU12, were significant for both positive, $t_{7750} = 4.10$, $p < .001$, $\eta_p^2 = .002$, and negative primate expressions $t_{7750} = 2.09$, $p = .037$, $\eta_p^2 = .001$. As expected, neutral expressions did not elicit patterned facial responses, $t_{7750} = 0.35$, $p = .725$, $\eta_p^2 = .000$ (for more details and a table with the summary statistics see S1 File).

## Moderation of mimicry by perceived emotion and self-reported closeness

Finally, we assessed whether, as shown for human-human mimicry, the degree of imitation was moderated by both the level of self-reported closeness and emotion perception [16,39,40]. For this, we calculated a positive pattern score based on the above mentioned Helmert contrast [35]. This score is positive when participants show a positive expression and negative when they show a negative expression. We conducted an analysis predicting the positive pattern score from both the rated positivity of the expression and self-reported closeness to the primate, predictors were z-scored (for the means and SDs for these variables see S1 File).

The model was significant $F(3, 2164) = 28.52$, $p = < 0.001$, $\eta_p^2 = 0.038$, with a significant effect of the rated positivity of the expression, $\beta = .13$, $SE = .02$, $t = 5.58$, $p < .001$, $\eta_p^2 = 0.014$, and a significant interaction between rated positivity and self-reported closeness, $\beta = .05$, $SE = .02$, $t = 2.45$, $p = .014$, $\eta_p^2 = 0.003$. Fig 4 shows the slopes.

Post-hoc tests confirmed that positive facial reactions of the observer increased with increased rated positivity at all levels of self-reported closeness. That is, the more positive the primate expressions were rated, the more positive was the participants' own expression. A reversal in sign for expressions rated not positive at all, suggests that participants showed negative expressions in response. Notably, for expressions rated more positively, the effect was moderated by perceived closeness such that the slope was steeper when participants reported feeling closer to the primate, $t_{2164} = 2.46$, $p = .038$ (for more details see S1 File).

## Discussion

This research is the first to provide evidence that humans not only recognize positive and negative emotion expressions shown by non-human primates but also spontaneously mimic these expressions. Further, as for human-human mimicry, the strength of participants' mimicry reaction depended on both their perception of the primate expression and their perceived closeness to the primate. These data suggest that humans spontaneously react empathically towards non-human primates.

Although there is evidence of yawn contagion between humans and a variety of species [41], the important difference between mimicry of emotional expressions and mimicry of facial gestures such as yawning is that the latter are not

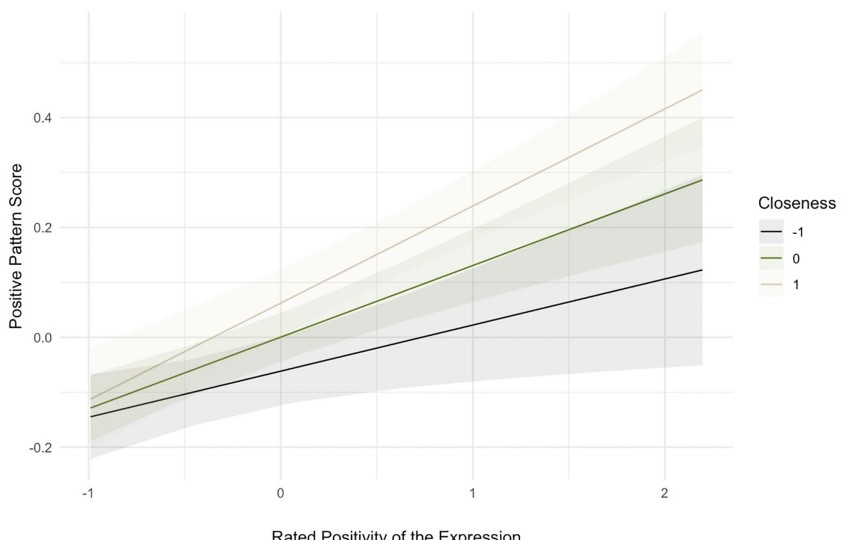

**Fig 4. Predicted values for expressive contrast as a function of rated positivity of the expression and self-reported closeness to the expresser.**

intrinsically meaningful -- they tell us little about the emotional state of the yawner. By contrast, in humans – and other primates -- emotion expressions signal socially relevant information [42] as well as behavioral intentions [43]. Specifically, whereas happiness signals affiliative intentions, anger signals the opposite. Hence, the mimicry of emotional expressions is linked to the capacity to understand the expresser and to empathically feel with the expresser [11] in a way that yawn contagion is not. Thus, the finding that humans mimic emotional behavior shown by primates sheds new light on the deep-rooted, cross-species nature of emotional connection between these different species and humans, suggesting that the human ability to empathize and mirror emotions extends beyond humankind.

What is also striking is that participants were not only able to identify the expressions in terms of negativity and positivity but also to correctly attach discrete emotion labels to the expressions. The positive expressions were play-faces, whereas the negative expressions were open-mouth threat displays, which participants rated as predominantly happy and angry respectively. These ratings represent a plausible interpretation of the behaviors and the accompanying behavioral intentions. Notably, both behaviors involve an open mouth with teeth visible. Nonetheless, participants were able to distinguish the behavioral intentions behind these facial gestures by attributing an affiliative vs antagonistic discrete emotion to the primates.

Finally, a regression analysis showed that the intensity of the mimicry behavior depended both on the perceived emotionality of the expression and the perceived closeness to the primate. Specifically, participants reported both more liking and more felt closeness for primates who showed positive expressions. In turn, mimicry of positive expressions was more pronounced when participants reported higher levels of felt closeness to the specific animal. Thus, the affiliative stance towards the primate is important for situations were mimicry produces an affiliative expression in the mimicker. That participants modulate positive mimicry as a function of the affiliation and closeness they feel towards the expresser suggests that people may in some ways be more "careful" when sending an affiliative signal, a smile, in response to a play-face, than when sending an essentially antagonistic signal, a frown, in response to a threat display. This more nuanced reaction to a positive, affiliative signal matches findings for human-human mimicry [16,40]. One could speculate that positive overtures towards others may be considered "costly" in some contexts and hence in those contexts should be preferentially shown to those we already like and feel close to. The question why mimicry of positive expression seems to be preferentially moderated by perceived closeness is intriguing and should be addressed by future research.

The present study provides strong evidence that humans not only understand positive, negative and neutral expressions shown by non-human primates but mimic these as well. It should be noted that the primates shown included not only the four great apes but also two different monkeys. As such, the findings allow for some generalization across primates.

Yet, the study also has some limitations. The videos were very short and showed not only the faces of the primates but also different levels of body postures. Also, the primates showing emotion expressions moved more than the ones showing a neutral expression. This notably lower level of arousal may have contributed to the use of the label sad for neutral expressions.

However, adding body cues may have helped participants to decode the expressions better. Further, participants were not only able to distinguish the positive and negative expressions from neutral expressions, but also from each other, despite the fact that both were associated with higher arousal and involved an open mouth that revealed the teeth.

In conclusion, the intricate relationship between emotional sharing and key dimensions of empathy suggests that our findings may transcend the boundaries of evolutionary biology, resonating deeply within psychology and the humanities. In particular, philosophy of mind, where foundational theoretical models of empathy have been developed, stands to be significantly influenced. If humans are capable of perceiving and resonating with the emotional states of non-human animals, this challenges long-standing anthropocentric paradigms and fosters a reconceptualization of the human-animal relationship.

The implications of this perspective extend beyond theoretical discourse, bearing profound ecological and ethical consequences. By reducing the psychological and conceptual distance between humans and other animals, we cultivate

a biocentric rather than anthropocentric worldview—one that acknowledges the intrinsic value of all living beings, independent of their utility to humankind. Such a shift is particularly crucial in addressing pressing global challenges, including biodiversity loss, the degradation of ecosystem services, and the destruction of natural habitats. Recognizing our shared emotional landscape with non-human animals can serve as a catalyst for fostering a deeper ecological sensitivity, inspiring policies and conservation strategies rooted in an ethic of care and interconnectedness.

## Supporting information

**S1 File. R Markdown.** This is the markdown for the R analyses.
(PDF)

## Author contributions

**Conceptualization:** Ursula Hess, Till Kastendieck, Elisabetta Palagi.

**Formal analysis:** Ursula Hess, Till Kastendieck.

**Methodology:** Ursula Hess, Till Kastendieck, Heidi Mauersberger, Marina Davila-Ross, Katja Liebal, Elisabetta Palagi.

**Project administration:** Till Kastendieck, M. Gizem Erkol.

**Supervision:** Ursula Hess, Heidi Mauersberger.

**Writing – original draft:** Ursula Hess, Till Kastendieck.

**Writing – review & editing:** Ursula Hess, Till Kastendieck, Heidi Mauersberger, Marina Davila-Ross, Katja Liebal, Elisabetta Palagi.

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
