## [Decision Letter · Decision Letter 0]

30 Dec 2025

Dear Dr. Hess,

Thank you for submitting your manuscript to PLOS ONE. After careful consideration, we feel that it has merit but does not fully meet PLOS ONE’s publication criteria as it currently stands. Therefore, we invite you to submit a revised version of the manuscript that addresses the points raised during the review process.

We look forward to receiving your revised manuscript.

Kind regards,

Brittany N. Florkiewicz, Ph.D.

Academic Editor

PLOS One

Journal Requirements:

2. Please note that your Data Availability Statement is currently missing the repository name. If your manuscript is accepted for publication, you will be asked to provide these details on a very short timeline. We therefore suggest that you provide this information now, though we will not hold up the peer review process if you are unable.

4. We note that Figure 1 in your submission contain copyrighted images. All PLOS content is published under the Creative Commons Attribution License (CC BY 4.0), which means that the manuscript, images, and Supporting Information files will be freely available online, and any third party is permitted to access, download, copy, distribute, and use these materials in any way, even commercially, with proper attribution. For more information, see our copyright guidelines: http://journals.plos.org/plosone/s/licenses-and-copyright.

1) You may seek permission from the original copyright holder of Figure 1 to publish the content specifically under the CC BY 4.0 license.

2) If you are unable to obtain permission from the original copyright holder to publish these figures under the CC BY 4.0 license or if the copyright holder’s requirements are incompatible with the CC BY 4.0 license, please either i) remove the figure or ii) supply a replacement figure that complies with the CC BY 4.0 license. Please check copyright information on all replacement figures and update the figure caption with source information.

If applicable, please specify in the figure caption text when a figure is similar but not identical to the original image and is therefore for illustrative purposes only.

**Additional Editor Comments:**

Thank you for submitting your manuscript to PLOS One! I apologize once again for the delay. I have now secured reviews from two qualified animal behaviorists. They both cite important theoretical and methodological concerns with your manuscript that should be fully addressed before your next submission. Below, you will find a copy of their comments.

Reviewers' comments:

Reviewer's Responses to Questions

**Comments to the Author**

1. Is the manuscript technically sound, and do the data support the conclusions?

Reviewer #1: Partly

Reviewer #2: Yes

2. Has the statistical analysis been performed appropriately and rigorously?

Reviewer #1: Yes

Reviewer #2: I Don't Know

3. Have the authors made all data underlying the findings in their manuscript fully available?

Reviewer #1: Yes

Reviewer #2: Yes

4. Is the manuscript presented in an intelligible fashion and written in standard English?

Reviewer #1: Yes

Reviewer #2: Yes

Reviewer #1: General

My assessment of the current paper is mainly focused on some aspects of the statistical analysis of the experimental protocol designed by the authors .

The authors investigated the combined effect of species (humans/ not-human primates) and task specificity in facial expressions (neutral, negative, positive), on conscious/ unconscious mimicry . and used minimally invasive protocols for data collection.

Standardized stimuli (2D videos of human facial expressions) and computerized data analysis (facial mimicry) showed a close connection between the emotional behavior of non human primates and that of humans. Even if the reported data and the relevant analysis and results are not totally new, further investigations may open new perspectives on the role of empathy inside evolution. Dynamic interactions between the varying facial expressions might also help communication inside and outside species, with possible interaction in the fields of human rehabilitation, as already done in various therapies with other mammals (horses, dogs..).

Statistics

I have some concern about the number of significant differences in the paper, a general effect of false positive inflation, with subsequent small sample sizes, potentially contributing to the reduced replication rate. Indeed, this topic is of current concern in several research contexts. For example, please see a recent meta analysis by Murphy et al, , Estimating the Replicability of Sports and Exercise Science Research. Sports Med. 2025 Oct;55(10):2659-2679. doi: 10.1007/s40279-025-02201-

Of course I do recognize that authors used several correction method estimators (post hoc tests, effect size, Cohen coefficients ...). But this is non-sufficient considering the limited number of landmarks/ variables used in the study, and the large numbers for statistical tests. From a mathematical point of view, an increase of statistical tests also increments the number of statistically significant values (p<0.05) but they are false positives. Which is the biological significance or practical/ clinical meaning of the current extra p values?

Reviewer #2: The paper presents interesting findings on cross-species mimicry of facial expressions from humans viewing primate stimuli; the paper generally is easy to read, and the findings are novel and interesting for the field, especially for comparative affective science. The paper requires revision, however, and adding more nuance. I made various comments on how the paper can be improved. These mainly refer to more careful reflections and assumptions, better embedding with the literature and more information in terms of the choice of statistics and descriptive stats, see below.

Before delving into each part of the paper, I would like to make a general remark, which concern the whole paper – and I believe we should all be mindful of as researchers especially when drawing assumptions of affective states in nonlinguistic species: What actually is an emotion expression in animals? Are all facial expressions in animals emotional in nature, and how do we know the expression we see is linked to the assumed affective state? Without physiology markers, what makes us sure that the expressions presented here do not comprise voluntary facial movements used for coordination (Waller et al. 2017, Neuroscience & Biobehavioral Reviews; Heesen et al., 2022 Primate Cognitive Studies)? We know certain primates have some voluntary power over their expressions (e.g., Waller et al., 2015 and several other papers). I think the field generally, including this novel paper, should apply more care in the way we phrase our work and how we use terms. Here, the authors might consider avoiding terms similar to “emotional mimicry” or “emotion expressions”, since they did not use physiological measures to support the analyses, neither for apes or humans.

Abstract

-There is a typo in the last phrase, I assume it could read as: …”suggesting that humans are able to emphasize and mirror emotions of other species.”

-I wondered: since empathy contains several “building blocks”, the term itself might not be adequate as you only looked at mimicry - which is a necessary part of it though not sufficient. A simple adjustment in phrasing is sufficient here.

Introduction

-“more egalitarian macaque species” – who are you comparing them to, to other macaque species? Please specify for clarity.

-P.4: Whether or not humans can classify primate expressions and how well they perform may be dependent on many more aspects than listed here; some studies using attention measures show they can (Kret & van Berlo, 2021) though it depends on age (children struggle at it); others using a priming design showed that humans do not pay selective attention to congruent facial expression matching a former affective scene, suggesting difficulty in cross-species perception also in adults (Heesen et al., 2024, PNAS Nexus); it may be helpful for readers if the authors better embed their current work with already existing literature on cross-species perception.

- P.4: I have a remark regarding the following phrase: “and hence an ability to categorize expressions into positive versus negative is plausible.” In primate behaviour, not every context may be as clear cut and thus separable on such a dichotomous scale – for example, contact-playing in primates or sexual contact in bonobos can be positive and negative at once (being close to a dominant and potentially aggressive other, while experiencing an arousing, affiliative contact); at the same time, mobbing group members or predators could combine positive and negative states at once (securing ones dominance status, dominating others, potentially being attacked by another). Such a dichomotous valence scale might be even difficult to apply to humans, because most interaction contexts are a messy pool of feelings with pleasant and unpleasant aspects – I would recommend caution in strong categorizations into positive and negative, especially when comparing across species. The authors may want to acknowledge that the phenomenon of valence is not as clear-cut as we’d like it to be, especially for animal researchers. Rather than calling expressions positive or negative, the field may benefit from more nuance moving forward, e.g., focusing on the functionality, or context in which expressions occur, rather than presuming whether the experience a good or bad state.

-P.5: Can the authors state more about which AUs they considered active in the threat related display of the primates? The display they show in the Fig. 1 looks like a mix between scream and bared teeth face, rather than open mouth threat face, see Figure 2 Parr et al 2007 (between bared teeth and scream face) and see Fig 1 c Waller et al 2016. Can the authors cite a paper that shows such a threat face expression as indicated in their Figure? Typically, threat face involves an open mouth, though teeth are not necessarily visible (see Fig 1 a and b, Overduin-de Vries et al 2016). If the expressions in the negative category were bared-teeth faces, then this needs to be discussed, because it would rather fit the “fear” than the “anger” category and it is interesting that people still attributed anger more so than fear (Fig 2, results).

Methods

-P.9: Did the authors instruct people not to talk, or have they verified this? They may have talked about the videos or said things while viewing, such that mouth movements could be affected by that – It would be good to verify.

-In the videos of the primates, were there any bystanders or objects visible? This should be mentioned somewhere.

-P. 9: The sentence was cut at the end of page.

-Could the others provide more information on what the test trial with the panthoot included?

-Did the authors ask participants whether they had any experience in watching apes, have they seen or observed them before? This is important as, like the authors mentioned, watching such videos could be affected by former training.

-Could the authors provide more information about the statistics chosen, models fitted, assumptions met within their LMMs? Use of software?

Results

-As indicated above, a summary statistics would be helpful for all ratings.

-Since the authors, in their discussion, generalize their findings across primate species, it would be beneficial to include a graph for each primate species here, such that the reader can get an insight whether the pattern holds equally across species, and across apes and monkeys in particular.

Discussion

-P.16 “That participants modulate positive mimicry as a function of the affiliation

and closeness they feel towards the expresser suggests that people may in some ways

be more “careful” when sending an affiliative signal, a smile, in response to a play-face,

than when sending an essentially antagonistic signal, a frown, in response to a threat

display.” – An alternative explanation could be that people feel more close, and therefore express more affiliative expressions. What do the authors mean exactly by “careful” and how is this explanation justified, i.e., do you have papers that support this idea, and what would it tell us if humans are more “careful” in affiliative contexts? In some way, one would expect the opposite: one should be most careful in negative contexts because those contexts bear danger and potential injuries. Also, why would you expect positive “overtures” (what is this?) to be more costly than coordinating in a potentially life-threatening conflict/fight? I would recommend adding some theoretical background and embedding the claims/conclusions/assumptions into the literature.

-At the end of the discussion, there is a random sentence on page 18 starting mid way, does it belong to the methods?

**Do you want your identity to be public for this peer review?** For information about this choice, including consent withdrawal, please see our Privacy Policy

Reviewer #1: No

Reviewer #2: No

---

## [Author Response · Author response to Decision Letter 1]

16 Jan 2026

Thank you for allowing us to revise our manuscript. We carefully read the reviewers’ comments and addressed them as outlined below.

We also moved the Ethics and Data availability statements to the end of the methods section. Depository names for the Data availability statement and the preregistration are included there.

The copy right for the images in Figure 1 is held by two of the authors and we have added this information to the figure caption and added a file entitled “Granted Permissions where the authors (Marina Davila-Ross and Katja Liebal) agree that the images included in the figure may be published under CC BY 4.0 license.

Review Comments to the Author

Reviewer #1:

1. General

My assessment of the current paper is mainly focused on some aspects of the statistical analysis of the experimental protocol designed by the authors.

The authors investigated the combined effect of species (humans/ not-human primates) and task specificity in facial expressions (neutral, negative, positive), on conscious/ unconscious mimicry. and used minimally invasive protocols for data collection.

Standardized stimuli (2D videos of human facial expressions) and computerized data analysis (facial mimicry) showed a close connection between the emotional behavior of non human primates and that of humans. Even if the reported data and the relevant analysis and results are not totally new, further investigations may open new perspectives on the role of empathy inside evolution. Dynamic interactions between the varying facial expressions might also help communication inside and outside species, with possible interaction in the fields of human rehabilitation, as already done in various therapies with other mammals (horses, dogs..).

The description of the study is not quite accurate as the stimuli consisted of short videos showing only primate expressions. Specifically, the goal of the study was to assess whether human observers can a) recognize the valence of primate expressions and b) mimic the expressions in terms of valence. We found both to be the case. The design did not allow for dynamic interactions strictly speaking.

2. Statistics

I have some concern about the number of significant differences in the paper, a general effect of false positive inflation, with subsequent small sample sizes, potentially contributing to the reduced replication rate. Indeed, this topic is of current concern in several research contexts. For example, please see a recent meta analysis by Murphy et al, , Estimating the Replicability of Sports and Exercise Science Research. Sports Med. 2025 Oct;55(10):2659-2679. doi: 10.1007/s40279-025-02201- Of course I do recognize that authors used several correction method estimators (post hoc tests, effect size, Cohen coefficients ...). But this is non-sufficient considering the limited number of landmarks/ variables used in the study, and the large numbers for statistical tests. From a mathematical point of view, an increase of statistical tests also increments the number of statistically significant values (p<0.05) but they are false positives. Which is the biological significance or practical/ clinical meaning of the current extra p values?

We share the reviewers concerns regarding power and sample size. We included a power simulation that was based on online studies of human mimicry of other humans but also avatars conducted in our laboratory. We started out with the mean effect size from that body of research and the power simulation suggested that N > 200 would allow for 90% power. We collected data from 230 participants. Data from 18 participants were excluded due to technical problems or noncompliance with the video instructions (e.g., eating during the experiment, face not visible on camera), thus we retained 212 participants. The power curve for the simulation was included in the supplementary materials.

As regards the statistical tests, the focal tests reported in the manuscript are the LMM equivalent of two one-way analyses of variance with the factor expression type for the negativity/positivity rating scales with associated alpha corrected post-hoc tests (Tukey) and a two-way analysis with AU and expression type. The latter was followed up by a-priori Helmert contrasts by expression type. A further correlational analysis assessed whether the Helmert contrast score varied as a function of the rated positivity of the expression and the self-reported closeness to the primate with associated post-hoc tests for the significant interaction. These are the minimal tests needed to assess the hypotheses. Hence cherry picking from an excessive array of tests was not possible. Note that all significant F-values were significant at p < .001 (as well as most post-hoc tests), suggesting stable effects. Post-hoc tests used family wise alpha protection via the Tukey test. We also report semi-partial eta2, allowing the reader to judge the size of the effect.

Given that the power simulation suggests that we used an adequate sample size and the sparse testing, we feel that our statistical analyses are commensurate with the reviewer’s desire for adequate power and stringent statistical tests.

Reviewer #2:

1. The paper presents interesting findings on cross-species mimicry of facial expressions from humans viewing primate stimuli; the paper generally is easy to read, and the findings are novel and interesting for the field, especially for comparative affective science. The paper requires revision, however, and adding more nuance. I made various comments on how the paper can be improved. These mainly refer to more careful reflections and assumptions, better embedding with the literature and more information in terms of the choice of statistics and descriptive stats, see below.

We thank the reviewer for these encouraging words. We found the suggestions helpful and have tried to implement them accordingly.

2. Before delving into each part of the paper, I would like to make a general remark, which concern the whole paper – and I believe we should all be mindful of as researchers especially when drawing assumptions of affective states in nonlinguistic species: What actually is an emotion expression in animals? Are all facial expressions in animals emotional in nature, and how do we know the expression we see is linked to the assumed affective state? Without physiology markers, what makes us sure that the expressions presented here do not comprise voluntary facial movements used for coordination (Waller et al. 2017, Neuroscience & Biobehavioral Reviews; Heesen et al., 2022 Primate Cognitive Studies)? We know certain primates have some voluntary power over their expressions (e.g., Waller et al., 2015 and several other papers). I think the field generally, including this novel paper, should apply more care in the way we phrase our work and how we use terms. Here, the authors might consider avoiding terms similar to “emotional mimicry” or “emotion expressions”, since they did not use physiological measures to support the analyses, neither for apes or humans.

We agree that we cannot know what the animal ‘feels’ – just as we cannot really know what humans feel. We use the term emotional mimicry because we take an observer focused stance in this research and in this case, we focus on the emotions that the (human) observers attribute to the primate. As is the case in emotional mimicry research in the human-human context, this is an attribution and it may be wrong. Nonetheless, observers act in accordance with these attributions. Hence, we opted to keep the terms emotional mimicry and emotional expression. However, we have added the paragraph below to explain our research perspective and to nuance these terms accordingly.

In this context it is important to emphasize that we use an observer focused perspective when using the terms emotional mimicry and emotional expression. It is difficult if not impossible to assert what any organism, humans and other primates included, experiences when showing a given expression. However, humans nonetheless not only attribute emotions to such expressions (18) but also act accordingly (19). Thus, to the human observer the animal shows an emotion and the observer will mimic (or not) the expression in line with this assumption. Thus, human empathy with a primate can only be based on the human observers‘ attributions, not on ground truth knowledge about inner states.

3. Abstract

-There is a typo in the last phrase, I assume it could read as: …”suggesting that humans are able to emphasize and mirror emotions of other species.”

We thank the reviewer for catching this typo

4. -I wondered: since empathy contains several “building blocks”, the term itself might not be adequate as you only looked at mimicry - which is a necessary part of it though not sufficient. A simple adjustment in phrasing is sufficient here.

Given the word limitations for the abstract, we added this important point to the introduction.

Specifically, mimicry is indicative of an affiliative stance towards the mimickee (1, 13) and often considered a “low road” to empathy (15), as such it is one important facet of human empathy.

5. Introduction

-“more egalitarian macaque species” – who are you comparing them to, to other macaque species? Please specify for clarity.

We thank the reviewer for the comment. We have revised this passage of the text, also in light of recent research on facial mimicry in other macaque species.

6. -P.4: Whether or not humans can classify primate expressions and how well they perform may be dependent on many more aspects than listed here; some studies using attention measures show they can (Kret & van Berlo, 2021) though it depends on age (children struggle at it); others using a priming design showed that humans do not pay selective attention to congruent facial expression matching a former affective scene, suggesting difficulty in cross-species perception also in adults (Heesen et al., 2024, PNAS Nexus); it may be helpful for readers if the authors better embed their current work with already existing literature on cross-species perception.

We were aware of this research; however, we originally decided to not include it in our literature review which specifically focused on studies where participants had to label the expressions. In essence these two studies present opposing findings based on different tasks which both differ from our task and in the case of Heesen et al. are also based on a very small sample. Nonetheless, given the sparsity of any research involving the perception of primate expressions by humans we now include these studies in the literature review

This notion is also supported by a study showing that adult humans rate complex scenes with multiple protagonists who are either human or bonobos similarly with regard to the valence and arousal of the picture. Even though the focus here was not on individual facial expressions per se, the expressive behavior of the bonobos played a role, as children tended to misidentify pictures including silent bared-teeth displays as positive (29). Yet, in a task were attentional bias to emotional versus neutral expressions was assessed the bias shown for human expressions was not found for images showing chimpanzee expression suggesting a lack of relevant expertise.

7. P.4: I have a remark regarding the following phrase: “and hence an ability to categorize expressions into positive versus negative is plausible.” In primate behaviour, not every context may be as clear cut and thus separable on such a dichotomous scale – for example, contact-playing in primates or sexual contact in bonobos can be positive and negative at once (being close to a dominant and potentially aggressive other, while experiencing an arousing, affiliative contact); at the same time, mobbing group members or predators could combine positive and negative states at once (securing ones dominance status, dominating others, potentially being attacked by another). Such a dichomotous valence scale might be even difficult to apply to humans, because most interaction contexts are a messy pool of feelings with pleasant and unpleasant aspects – I would recommend caution in strong categorizations into positive and negative, especially when comparing across species. The authors may want to acknowledge that the phenomenon of valence is not as clear-cut as we’d like it to be, especially for animal researchers. Rather than calling expressions positive or negative, the field may benefit from more nuance moving forward, e.g., focusing on the functionality, or context in which expressions occur, rather than presuming whether the experience a good or bad state.

We very much agree with this view. This is why we did not ask participants to classify expressions as either positive or negative but to indicate the degree of positivity and negativity of the expression on two separate dimensional scales. This allows for the possibility to describe expressions as both positive and negative to some degree. To avoid any confusion in that regard we rephrased the sentence to read:

and hence an ability to rate expressions as positive or negative is plausible.

8. -P.5: Can the authors state more about which AUs they considered active in the threat related display of the primates? The display they show in the Fig. 1 looks like a mix between scream and bared teeth face, rather than open mouth threat face, see Figure 2 Parr et al 2007 (between bared teeth and scream face) and see Fig 1 c Waller et al 2016. Can the authors cite a paper that shows such a threat face expression as indicated in their Figure? Typically, threat face involves an open mouth, though teeth are not necessarily visible (see Fig 1 a and b, Overduin-de Vries et al 2016). If the expressions in the negative category were bared-teeth faces, then this needs to be discussed, because it would rather fit the “fear” than the “anger” category and it is interesting that people still attributed anger more so than fear (Fig 2, results).

We did not use FACS to code the expressions. Rather, the expressions were independently rated by two of the authors who are experts in primate expressions, Elisabetta Palagi and Marina Davila-Ross, with regard to the expression shown. From these we selected five expressions for each category where the raters agreed. The participant ratings support the notion that indeed threat faces and not silent bared teeth displays were selected for the negative category. Importantly, however, the raters – like the participants – rated the whole video sequence.

9. Methods

-P.9: Did the authors instruct people not to talk, or have they verified this? They may have talked about the videos or said things while viewing, such that mouth movements could be affected by that – It would be good to verify.

As noted in the method section, participants who did not comply with instructions were excluded from the study. This included activities like eating but also taking or chewing gum.

10. -In the videos of the primates, were there any bystanders or objects visible? This should be mentioned somewhere.

We note in the introduction that the videos showed the animals in natural surroundings. We further note in the method section that only a single individual was shown in each video.

11. -P. 9: The sentence was cut at the end of page.

We thank the reviewer for catching this problem.

12. -Could the others provide more information on what the test trial with the panthoot included?

This was simply a single trial to familiarize the participants with the procedure. This is now specified.

13. -Did the authors ask participants whether they had any experience in watching apes, have they seen or observed them before? This is important as, like the authors mentioned, watching such videos could be affected by former training.

We did not ask this question. However, given the participant pool (Prolific academic) such knowledge would have been rare.

14. -Could the authors provide more information about the statistics chosen, models fitted, assumptions met within their LMMs? Use of software?

A markdown for the analyses is included in the supplementary materials. However, for ease of refere

---

## [Editor Report · Decision Letter 1]

19 Jan 2026

Evolutionary Echoes of Emotion: Humans Mimic Other Primate Expressions

PONE-D-25-31307R1

Dear Dr. Hess,

We’re pleased to inform you that your manuscript has been judged scientifically suitable for publication. Thank you for thoroughly addressing the reviewers' comments and concerns. Your article will be formally accepted for publication once it meets all outstanding technical requirements.

Kind regards,

Brittany N. Florkiewicz, Ph.D.

Academic Editor

PLOS One

Additional Editor Comments (optional):

Thank you for diligently addressing all of the reviewers' comments and concerns! After reviewing your revisions, I believe you have sufficiently addressed everything. However, there is one minor issue that can be resolved during the proofing stage: the quality of the images of the example NHP facial signals is relatively low on our end. Please ensure that these images are of the highest quality when you proofread your article.